# Significance of the Work Environment and Personal Resources for Employees’ Well-Being at Work in the Hospitality Sector

**DOI:** 10.3390/ijerph192316165

**Published:** 2022-12-02

**Authors:** Dunja Demirović Bajrami, Marko D. Petrović, Dejan Sekulić, Milan M. Radovanović, Ivana Blešić, Nikola Vuksanović, Marija Cimbaljević, Tatiana N. Tretiakova

**Affiliations:** 1Geographical Institute « Jovan Cvijić« Serbian Academy of Sciences and Arts, 11000 Belgrade, Serbia; 2Institute of Sports, Tourism and Service, South Ural State University, 454080 Chelyabinsk, Russia; 3Faculty of Hotel Management and Tourism, University of Kragujevac, 36210 Vrnjačka Banja, Serbia; 4Department of Geography, Tourism and Hotel Management, Faculty of Sciences, University of Novi Sad, 21000 Novi Sad, Serbia; 5Faculty of Management, University Union “Nikola Tesla”, 21205 Sremski Karlovci, Serbia

**Keywords:** job demand, job control, social support, personal resources, employees’ well-being at work, hospitality

## Abstract

The aim of the study was to investigate whether different elements of the work environment (manifested by job demands, job control, and social support) and personal resources were linked to employees’ well-being at work. Based on data gathered from 574 employees in the hospitality industry in Serbia, it was also tested if personal resources, expressed through self-efficacy, hope, optimism, and resilience, could moderate the relationship between work environment and employees’ well-being at work. Correlation analyses showed that high job demands had negative effects on employees’ well-being, causing negative emotional reactions to their job, while job control and social support developed positive relationships with positive employees’ well-being. The moderating effect analysis found that personal resources can fully moderate the relationship between job demands and well-being at work, and job control and well-being at work. On the other side, personal resources were not a significant moderator in the relationship between social support and well-being at work, indicating that even when employees have adequate personal resources, they are not enough to decrease the negative effects of lack of social support on employees’ well-being at work. This shows how important the support of supervisors and colleagues is for employees in hospitality.

## 1. Introduction

There is a general agreement between researchers and different organizations that employees’ well-being and health and their preservation are of essential importance for the global market. Well-being at work (or workplace well-being) was defined by the International Labor Organization as “all aspects of working life, from the quality and safety of the physical environment, to how workers feel about their work, their working environment, the climate at work and work organization” [1]. A higher level of well-being influences employees to perform better since it is believed that a “happy worker is a productive worker” [2]. High and stressful job demands and low ability to control the conditions under which the work is completed can significantly predict the level of employees’ well-being [3], while reduced well-being can be expressed through the appearance of work-related anxiety, stress, depression, fatigue and other undesirable states and emotions [4,5]. In addition to conditions of the work environment, a reduced level of well-being can be caused by a deficiency in the level of employees’ personal resources which they need to cope with challenges more easily in the workplace [6]. On contrary to this, it was evident that when employees receive adequate support in their work environment, it can boost their personal resources, such as self-efficacy and positive affectivity and, in turn, decrease turnover intentions and encourage employees to devote themselves to clients and solving their problems [7]. Stimulating conditions of the work environment and higher levels of personal resources can eliminate the negative effects of burnout and encourage the development of positive emotions that a job can make a person feel [8]. Employees’ well-being at work may impact their behavior at work and their relationships with colleagues and supervisors, employees’ productivity and performances, and turnover intentions, but it can also impact customer loyalty [9,10,11]. When the workplace and job cause positive emotions, it will reflect on both employees and organizational outcomes [2].

The hospitality industry is a labor-intensive industry since it requires comprehensive employment of labor, so employees are one of the most important factors in achieving the competitiveness of the sector. Although employees have an important role, the hospitality industry is known as one of the branches with the highest rate of employee turnover, while managers are facing difficulties retaining employees [12,13]. Reasons for high rates of turnover can be found in high job demands, manifested by high seasonality and, therefore, job insecurity, bad working conditions, and monotonous jobs that are constantly repeated [14,15]. Additionally, many employees do not have enough job control and the lack of supervisor support decreases the sense of belonging to the organization and their well-being [15]. Finding ways to increase the competitiveness of the hospitality industry by reducing turnover and increasing employees’ well-being has become the primary focus of both managers and researchers. Of all available models for examining the impact of various factors on the well-being of employees, the job demand-control-support model [16] was recognized as the most suitable for determining the impact of the work environment on preserving employees’ well-being [17]. Although this model can provide valuable insights into ways to preserve employees’ well-being at work, it has not been used much in the context of tourism and hospitality research [18,19,20]. 

Grounded on all the points mentioned above, the purpose of our study was to investigate whether different elements of the work environment (manifested by job demands, job control, and social support) and personal resources were linked to employees’ well-being at work. Based on data gathered from employees of the hospitality industry in Serbia, we also tested the moderating role of personal resources, expressed through self-efficacy, hope, optimism, and resilience, in the relationship between the work environment and employees’ well-being at work. 

### 1.1. Relationship between the Work Environment and Employees’ Well-Being at Work

The work environment has an important impact on employees’ well-being since employees spend considerable amounts of time at work during their lives. Previous studies indicated that elements of the work environment such as job demands, job control, and social support significantly impacted employees’ well-being and other work-related outcomes [21,22,23]. Karasek Jr. [24] developed a model where these three elements were integrated into the so-called job demand-control-support (JDCS) model. The model suggested that if employees are faced with high job demands and low job control and low social support, it will have a negative impact on the well-being of employees, causing, for example, psychological strain, depression, and job-related anxiety. The JDCS model was widely used to analyze the relationship between work environment characteristics and well-being [17,25], burnout [21,26], work engagement [27,28], and job satisfaction [29].

Job demands were defined as different aspects of a job (e.g., physical, psychological, and organizational), that require physical and psychological efforts for their accomplishment [30]. Work pressure, long work hours, work overload, inadequate professional demands, and role conflicts are all synonyms for high job demands. Previously, it was highlighted that when job demands are not adequate, they can exhaust employees’ resources and trigger undesirable states, such as health and well-being issues [9]. Further, employees’ health and well-being issues caused by job demands can develop cynical attitudes towards the job that can impact employees’ performances [31]. As stated by Lawson, Noblet, and Rodwell [22], job demands can be used as a predictor for psychological health and job satisfaction, but some authors believe that the negative impact of job demands on employees’ well-being does not have to happen by default, because they will act as stressors only when a high effort is necessary to meet working requirements [32]. High job control, reflected through work-related autonomy, is positively associated with employees’ well-being, leading to increased work engagement and positive feelings, such as happiness [33]. Additionally, higher job control can encourage employees to learn new things because they will perceive their work as more challenging [25]. Research by Joudrey and Wallace [34] and Nerobkova et al. [35] revealed that having control over deciding how many hours to work can decrease depression, while employees will feel more empowered if their tasks are clearly defined [36]. Social support, presented through colleague and supervisor support, is a significant stimulus for employees’ development and well-being [25,36]. The possibility to rely on colleagues in a personal and emotional way improved employees’ psychological well-being, but a lack of empathy among colleagues facing the same high demands in the workplace leads to a lack of mutual support [34]. 

In the tourism and hospitality industry, employees are facing high job demands such as monotonous jobs, low job security, long working hours, night and weekend work shifts, poor work equipment, and others [37]. Even some authors [38] indicated that the job of certain employees in hospitality, such as waiters and cooks, can be classified as a high-strain job because it is characterized by high psychological and physical demands and low job control. These conditions have a strong negative impact on the health and well-being of employees. Research on the impact of different elements of the work environment on hospitality was mostly completed on a sample of hotel workers. On a sample of hotel managers in the USA, O’Neill and Xiao [39] showed that job demands were positively correlated with employees’ emotional exhaustion, while research completed on hotel employees from South Korea showed that social support and encouragement of self-esteem positively influenced employees occupational and subjective well-being [40]. Based on everything mentioned above, we proposed the following hypotheses:

**H1a:** 
*High job demand, as an element of the work environment, will have a negative impact on employees’ well-being at work.*


**H1b:** 
*High job control, as an element of the work environment, will have a positive impact on employees’ well-being at work.*


**H1c:** 
*High social support from colleagues and supervisors, as an element of the work environment, will have a positive impact on employees’ well-being at work.*


### 1.2. Relationship between Employees’ Personal Resources and Their Well-Being at Work

Personal resources are described as an individual’s assets that are at a person’s disposal to enhance the effective functioning of some work or life domains and include a sense of ability to control and impact those domains successfully [41,42,43]. Personal resources are valuable assets that help in overcoming stressful situations more easily and in fulfilling the set goals, especially those with hindrances [6]. It is determined that individuals who are high in assertiveness, resilience, and self-efficacy can better cope with unexpected events, stand up for themselves or even recover easily from failure, distress, or frustration [44]. In our research, we focused on four personal resources that were most represented in previous research—hope, resilience, self-efficacy, and optimism (known as psychological capital). The importance of personal resources for someone’s well-being was recognized in previous research. Personal resources were associated with stress reduction, higher well-being, lower level of burnout, higher work engagement, and lower level of turnover intention. In the research completed by Reis, Hoppe, and Schroder [45], personal resources were marked as positive resources that can enhance employees’ psychological well-being, and the positive effects of these resources can even persist over time. Personal attributes, such as resilience, have strong motivational effects on well-being and can significantly boost it [6,36,46]. Research completed on a sample of Spanish teachers revealed that based on a level of personal resources (e.g., emotional skills) at the beginning of the academic year, school management can even predict a level of teachers’ exhaustion, cynicism, and depersonalization at the end of that year [47]. Enthusiastic employees seem to develop positive emotions more easily, such as happiness or enjoyment, and have better mental health [48]. With those positive states, employees can influence their colleagues to increase their work engagement and feel better. Employees with a strong sense of resilience, optimism, hope, and self-efficacy are more vigorous when doing their jobs and can develop positive feelings toward their workplaces [8]. This indicates that personal resources can act as an independent predictor of employees’ well-being (e.g., work engagement). In the hospitality sector, personal resources were linked to emotional exhaustion and turnover intentions of hotel workers in Cameroon [7], while a negative correlation between self-efficacy and emotional exhaustion was determined in a sample of hotel workers in Nigeria [49]. In the research of Paek et al. [50], it was once again confirmed that personal resources (i.e., psychological capital) are a strong predictor of work engagement for frontline staff in Korea’s hotels, but they can also act as a predictor of employees’ morale variables (affective organizational commitment and job satisfaction). Although many studies investigated the influence of employees’ personal resources on various outcomes in their workplaces, it is still insufficiently researched how personal resources relate to people’s emotional reactions to their job. Based on this, the following hypothesis is set: 

**H2:** 
*Employees’ personal resources (hope, resilience, self-efficacy, and optimism) will have a positive impact on employees’ well-being at work.*


### 1.3. Moderating Role of Employees’ Personal Resources in the Relationship between Work Environment and Employees’ Well-Being at Work 

In addition to direct resources, personal resources can act as a moderator between work environment and job-related outcomes, such as burnout or engagement, but for this moderating effect, there is no total agreement in the previous studies. Self-efficacy helped in decreasing the negative effects of a low supporting climate in the hospitality industry on employees’ performances and intention to quit the job [50]. Additionally, personal resilience moderated the negative effects of emotion-oriented coping on depression for undergraduates [51]. The results of research completed by Karatepe [7] revealed that positive affectivity, intrinsic motivation, and self-efficacy moderated the effect of perceived organizational support on emotional exhaustion, extra-role customer service, and turnover intentions of frontline hotel employees in Cameroon. In the healthcare sector, personal resources expressed through emotion regulation strategies reduced the negative effects of high job demands on employees’ well-being and decreased the level of stress, but only if employees went through standardized emotion regulation training [6]. Contrary to this, Xanthopoulou et al. [52] found that personal resources reflected through self-efficacy, organizational-based self-esteem, and optimism did not moderate the relationship between job demands and exhaustion of Dutch employees. As further research on the moderating role of personal resources is necessary, the following hypothesis was put forward: 

**H3a:** 
*Personal resources (hope, resilience, self-efficacy, and optimism) will moderate the negative effects of high job demands on employees’ well-being at work.*


**H3b:** 
*Personal resources (hope, resilience, self-efficacy, and optimism) will moderate the negative effects of low job control on employees’ well-being at work.*


**H3c:** 
*Personal resources (hope, resilience, self-efficacy, and optimism) will moderate the negative effects of low social support on employees’ well-being at work.*


### 1.4. Control Variables 

Previous similar research provided evidence that demographic characteristics such as gender, marital status, and years of work experience were important dependent variables. Numerous studies have shown that female employees experienced depression and reported more stress compared to men when they are exposed to high job demands [53,54,55]. When employees are faced with high job demands, low control, and low social support, female managers reported more psychosomatic complaints than male managers causing poorer health in female employees [56]. Female employees felt more exhausted compared to men, and women rated that they were faced more often with high demands and less control and support compared to men at their workplace [21,57]. When it comes to marital status, the findings were inconsistent. Zhang et al. [58] indicated that female employees and those with children had difficulties in coping with high demands at work because they were burdened with numerous demands at home as well, leading to their lower well-being. On the other side, Bhumika [59] pointed to the fact that married employees coped better with high demands at work when they had spousal support with household responsibilities, and this was especially emphasized during the COVID-19 pandemic when many employees worked from home. Although previous research found an impact of marital status and gender, there is the research of Kumar, Alok, and Banerjee [36], which showed that gender and marital status (employees with children) were not significant when facing high job demands since employees who worked from home during COVID-19 received enough support from household members. Years of work experience is a significant factor, since it was proved that those employees who have greater work experience can better adjust to a job’s expectations and follow its routine more easily [34]. Based on all presented above, the significance of gender, marital status, and years of work experience was taken into consideration when analyzing the impact of the work environment and employees’ personal resources on their well-being at work. 

## 2. Materials and Methods

### 2.1. Data Collection and Sample 

Study data were collected from employees who worked in hotels of all categories (frontline personnel), restaurants, and travel agencies located in the major tourist destinations in Serbia. These tourist destinations were marked by the Statistical Office of the Republic of Serbia as the most important since they recorded the highest number of overnight stays in 2021 [60]. The major destinations were spa resorts (2.6 million overnight stays), with Vrnjačka Banja and Sokobanja as the most frequently visited, then the mountain resorts Zlatibor and Kopaonik, with 1.9 million nights, and the third group were urban centers, with Belgrade (the Serbian capital city) as the most visited destination (1.6 million overnight stays). We included in the research only those tourism service providers that were registered at the Serbian Business Registers Agency. The sampling procedure was completed from February to April 2022. Managers of the selected hotels, restaurants, and travel agencies were invited, via email, to be part of the research; we sent a link to the survey to their frontline employees. In total, 574 questionnaires were collected via a web-based survey.

Based on the results of descriptive statistics, women were slightly more represented in the sample (54%) compared to men (46%). Most of the respondents belonged to two age groups—between 29 and 38 years (43%), and 39 and 48 years (39%). When it comes to education level, most of the respondents finished the first stage of tertiary education (Bachelor’s or Master’s level)—52%, or post-secondary non-tertiary education (45%). Among the respondents, 37% had between 6 and 10 years of work experience in tourism, and 32% had between 15 and 19 years of work experience. Regarding marital status, 44% were married and had kids, 29% were married without kids, 17% were single, and 10% were divorced. 

### 2.2. Measures 

The job content questionnaire (JCQ), developed by Karasek et al. [16] was used to measure the work environment expressed through three dimensions—demand, control, and social support. Job demand had five items and reflected roles and expectations of employees’ jobs. Cronbach’s alpha was 0.81 and the sample item was ”Do you have sufficient time for all your work tasks?“. A decision latitude or job control was measured with nine items showing employees’ capability to determine how and when to accomplish job assignments. A sample item was „Does your job require doing the same tasks over and over again?“, while Cronbach’s alpha was 0.87. The items were scored on a five-point scale ranging from 1 (never) to 5 (often). The last dimension, social support, had eight items divided into two indicators—colleague support (four items) and supervisor support (four items). Social support points to how one person can help another person to enhance his/her well-being by using interpersonal resources. Cronbach’s alpha was 0.84, while the sample item was “There is a good collegiality at work“. The respondents expressed their agreement or disagreement on a five-point scale ranging from 1 (strongly disagree) to 5 (strongly agree). 

Personal resources manifested through self-efficacy, optimism, hope, and psychological resilience were assessed with 24 items originating from Luthans et al.’s [61] theory. In our research, personal resources were viewed as a composite construct, since it was determined that the effects of components are lower if each component was considered individually [62]. Self-efficacy had seven items and shows how much employees are self-confident about their ability to face challenges. A sample item was “I feel confident when I am looking for a solution to a long-term problem”. Optimism relates to a positive attitude in terms of present and future and was measured with six items. The sample item was “If I were in a difficult situation at work, I could think of many ways to get out of it”. Hope, as a third dimension, was assessed with six items and it referred to the state of positive motivation that can lead to the accomplishment of set goals using different resources. The sample item was “At work, I am optimistic about what will happen in the future”. Finally, resilience contained seven items and reflects employees’ ability to recover from failure, distress, or frustration. The sample item was “In general, I can easily step over the more stressful things at work”. The respondents could answer the items using a five-point scale, ranging from 1 (strongly disagree) to 5 (strongly agree). Cronbach’s alpha values were 0.75 for self-efficacy, 0.71 for optimism, 0.69 for hope, and 0.77 for resilience. 

Employees’ well-being at work was analyzed with the job-related affective well-being scale developed by van Katwyk et al. [63], containing 20 items (shortened version), divided into two dimensions—positive affect and negative affect. Using a five-point scale (1—never, 5—always), employees indicated how any part of their job (e.g., work, coworkers, and supervisor) made them feel in the past 30 days, describing different emotions. The examples items were “My job made me feel bored“ (negative affect) and “My job made me feel inspired“ (positive affect). Cronbach’s alpha was 0.84 (positive affect) and 0.81 (negative affect). 

Demographic variables such as gender, marital status, and years of work experience were included as control variables. 

### 2.3. Data Analysis 

The statistical software package SPSS, version 26.0 (IBM, Armonk, NY, New York, NY, USA) was used to carry out descriptive statistics and correlation analyses. Confirmatory factorial analysis (CFA) was used to estimate the latent components measurement model and test it for construct validity and reliability. Cronbach’s alpha coefficients were used to assess the scale’s reliability. After checking the validity and reliability of each construct, structural equation modeling analyses (SEM analyses) were used to measure and analyze the relationships of observed variables. Descriptive statistics and correlation analyses were applied to better understand the differences between respondents and the background information of all variables.

Moreover, to check if the common method variance (CMV) was an issue in our research, we used a Harman one-factor analysis since, in the existing literature, this post hoc procedure was one of the most used to control CMV across all fields [64,65]. During this analysis, all items are loaded to confirmatory factor analysis with the aim of finding out if one single factor emerged. We entered all items of our variables into the SPSS file. The generated PCA output indicated that all items accounted for 58% of the total variance. The first unrotated factor captured only 26% of the variance in data. Since no single factor emerged and the first factor did not account for most of the covariance, we could conclude that CMV was not an issue in our research. 

## 3. Results

### 3.1. Correlation Analysis

Table 1 shows the results of the basic statistics (mean and standard deviation) and correlations between the main variables. Correlation analysis is used to measure the strength of the relationship between two variables, i.e., the level of change in one variable due to changes in the other variable. One of the main advantages of this statistical method is its practical simplicity and easiness of determining how the variables are related. If a result is positive, it means that both variables will increase in relation to each other, while a negative correlation refers to a conclusion that if one variable decreases, the other will increase. In our case, a correlation analysis confirmed significant negative relationships between job demands and employees’ well-being at work, and job demands and personal resources. Additionally, components of the work environment, job control, and social support were both positively correlated with well-being at work, while only social support positively correlated with personal resources. Moreover, personal resources were positively correlated with employees’ well-being at work. Finally, the control variables (gender, marital status, and years of work experience) showed that all control variables were correlated with all three components of the work environment (job demands, job control, and social support) and well-being at work. 

### 3.2. Results of the Path Model

In the next step, the model’s validity was examined. The results showed a good model fit since it had an acceptable level of goodness-of-fit statistics (*χ*^2^ = 2543.294, *df* = 1279, *p* < 0.001, *χ*^2^/*df* = 2.875, RMSEA = 0.046, CFI = 0.919, IFI = 0.95, TLI = 0.903). The proposed model had a relatively large level of prediction power of employees’ well-being at work since it accounted for 58.2% of the total variance. Further, the hypotheses were tested. Results are presented in Table 2 and Figure 1.

As expected, the results supported that job demands were negatively associated with employees’ well-being at work (β = −0.198, *p* < 0.001), supporting Hypothesis 1a. The other two components of the work environment, job control, and social support developed a positive correlation with employees’ well-being at work (β = 0.351, *p* < 0.001 and β = 0.426, *p* < 0.001, respectively), supporting Hypotheses 1b and 1c. Hypothesis 2 stated that employees’ personal resources can positively impact their well-being at work, and this was proved since this relationship was statistically significant (β = 0.411, *p* < 0.001). 

Table 2 also shows the moderating effect of employees’ personal resources. According to the results, personal resources fully moderated the relationship between job demands and well-being at work (β = −0.108, *p* < 0.001), and job control and well-being at work (β = 0.029, *p* < 0.001), but did not moderate the relationship between social support and well-being at work (β = 0.021, *p* > 0.05). Consequently, Hypotheses H3a and H3b were supported, while H3c was rejected. 

We also calculated a simple slope and constructed a regulation effect diagram that disclosed the moderating effect of personal resources in the relationship between the main constructs (work environment, job demand, job control and social support, and employees’ well-being at work). As shown in Figure 2, when the level of personal resources is high, it is more likely that high job demands will not have such a negative influence on employees’ well-being at work (*B* = 0.25, *t* = 3.580, *p* < 0.01). Additionally, a higher level of personal resources may moderate the relationship between job control (low level) and well-being at work (*B* = 0.26, *t* = 3.588, *p* < 0.01), showing that the reduced ability to control a job will not produce so many negative emotions towards a job if the employee has positive personal resources. Contrary to our expectations, Figure 2 showed that the level of personal resources (higher and lower) was not significant for positively correlated social support and well-being at work (*B* = 0.18, *t* = 3.459, *p* > 0.08), indicating that an even higher level of personal resources is not enough to reduce the negative effects of a lack of support on well-being at work. 

## 4. Discussion

The current study investigated whether different elements of the work environment in the hospitality sector (manifested by job demands, job control, and social support) and employees’ personal resources were linked to their well-being at work. Correlation analyses showed that all three components of the work environment were significantly associated with employees’ well-being at work. Specifically, high job demands had negative effects on employees’ well-being, causing negative emotional reactions to their job and revealing that job demands can strongly predict lower well-being of employees in hospitality. This was in line with some previous research [9,66,67,68] that found positive relationships between job demands, emotional exhaustion, and job-related anxiety. Regarding job control and social support, it was indicated that both components developed positive relationships with positive employees’ well-being, showing that greater work autonomy and the possibility to control some work conditions made employees feel enthusiastic, calm, and content. Additionally, supervisors’ and colleagues’ support has significantly contributed to improving the well-being of employees. These results are consistent with previous studies [69,70] showing that higher job control made employees feel more effective and that social support boosted employees’ well-being, but our results were different from results presented in research completed by Tims, Bakker, and Derks [71], who found social support had positive but insignificant relation to well-being. Although both work environment components (job control and social support) had a significant relationship with well-being, it was noticeable that the effect of the support on well-being was slightly stronger compared to job control which is in line with research completed by Asif, Javed, and Janjua [9], where the results indicated that social support had a more promising effect on anxiety than job control. 

Personal resources developed a significant positive relationship with employees’ well-being, suggesting that a high level of optimism, hope, resilience, and self-efficacy boosted positive emotions toward the job, that is, they can improve the well-being of employees. The importance of employees’ personal resources for their well-being at work was also highlighted in some studies confirming our results [7,8]. 

Inclusion of the employees’ demographic characteristics into the analysis showed that gender, marital status, and years of work experience were significant for a better and more complete understanding of employees’ well-being at work contrary to the research of Kumar, Alok, and Banerjee [36]. Thus, female employees in hospitality reported more negative emotions towards their job compared to men when they were exposed to less favorable working environment conditions (high job demands, low job control, and low social support). These findings were in line with previous research [58,59,60]. The opposite of the research by Bhumika [59], but supporting the research by Zhang et al. [58]. Employees with children had a lower level of well-being at work compared to single and married employees without children when they were faced with high job demands and low social support. This can be interpreted by the fact that employees having children are faced with additional demands at home which, together with the demands at work and the absence of support, makes them more exhausted and more susceptible to creating negative emotions towards the job. Employees with more working experience did not develop so many negative emotions compared to beginners when they were faced with job demands, which can be explained by the fact that they are more familiar with the expectations of the workplace (as stated by Joudrey and Wallace [34]). In contrast, more experienced employees were more sensitive to a lack of support from colleagues and supervisors. 

The moderating effect analysis found that personal resources can fully moderate the relationship between job demands and well-being at work, and job control and well-being at work. Specifically, the results guide us to conclusions that a high level of self-efficacy, optimism, hope, and resilience can mitigate the development of negative emotions toward jobs when employees face high job demands and a lower level of job control. Contrary to our expectations, personal resources were not a significant moderator in the relationship between social support and well-being at work, indicating that even when employees have adequate personal resources, they are not enough to decrease the negative effects of lack of social support on employees’ well-being at work. This shows how important the support of supervisors and colleagues is for employees in hospitality.

### 4.1. Practical and Theoretical Implications 

For the organizational policy in hospitality, this study implies that management should pay special attention to components of the work environment since it was shown that job demands, job control, and social support were significantly correlated with employees’ well-being at work and the state of their personal resources. High demands at work are one of the characteristics of hospitality (especially during the high season) and they often influence the creation of negative emotions towards work among employees and can even reduce the level of personal resources if they are present in the long term. To decrease the negative effects of high job demands, hospitality managers should set mutual expectations and clear priorities with their employees, make sure that employees are at the right workplaces according to their abilities and interests, enable task or even job rotation, and do their best to develop a work environment that will be challenging and unthreatening to employees’ well-being. Social support proved to be one of the most important factors that influenced employees’ well-being at work. In order to supervisors be supportive and boost employees’ well-being, they need to provide extended job autonomy for employees, organize work with clear goals and a reasonable timeframe for the completion of work-related tasks, interact with employees by giving them feedback and be specific, showing empathy and making the conversation “a two-way street”. This is especially necessary for the elderly and employees with families (children) because they have a harder time coping with excessive demands at work and a lack of support. In addition to this, some organizational interventions should be implemented such as giving new tasks to employees from time to time to avoid monotonous work, explaining how employees’ work benefits the organization, and creating incentive programs. Since the success of a hospitality organization depends on teamwork, some group and intergroup interventions are necessary (e.g., team building). It is known that supervisor support can activate learning among employees [25], so adequate and constant support can stimulate employees’ personal development and, therefore well-being at work. Employees who develop positive emotions toward their jobs will be more motivated and will put more effort into fulfilling their tasks. Additionally, they can motivate their colleagues and help them to easier overcome difficulties in the workplace. Increasing the level of personal resources can be achieved through different types of training (e.g., emotion regulation and stress-management). Providing support by increasing the level of employees’ personal resources and attracting new employees high in self-efficacy, hope, optimism, and resilience can indirectly impact their work engagement (in a positive direction) and turnover intention (by reducing the intention to leave the current organization) [72]. Finally, employees will see that the organization cares about their well-being at work only if the management initiates a good relationship between employees, their colleagues, and supervisors if they communicate regularly with employees and enable them to express their opinions and value and respect their efforts at the workplace. 

The findings of this study suggest that the job demand-control-support model can be successfully applied to the hospitality sector and can contribute to revealing which factors can act as stressors for employees’ well-being at work. It was evident that all three components of the work environment created a significant relationship with employees’ emotions towards the job. Additionally, the study enhances the existing knowledge about the impact of the work environment on personal resources, indicating that job control does not produce significant relationships. Further, the results of the joint effects of hope, optimism, self-efficacy, and resilience as personal resources on well-being at work are important since the hospitality literature is lacking that evidence. Eventually, when applying the JDCS model in hospitality, there was empirical support for the moderating role of personal resources, but not all indirect effects were of importance (for the relationship between social support and well-being at work). 

### 4.2. Study Limitations and Recommendations for Future Research 

This study is not without some limitations. First, when we started our research, we were concerned about the appearance of common method variance (CMV) since we planned to use a self-report questionnaire and collect data from the same participants for dependent and independent variables at the same time. The formation of CMV under these conditions has been recognized in the literature [73]. As suggested by Podsakoff et al. [65], some procedural remedies should be implemented to control common method variance. In our research, we implemented a few measures to reduce the possibility of its formation. When it comes to the questionnaire, we made the statements to be clear and concise to reduce the possibility of misunderstanding. Since the original scales were in English, when we translated them into Serbian, we tried to use familiar expressions adapted to the local context. It is known that complex and unclear syntax can lead to a large percentage of random responses [74]. Some scales had both positively and negatively worded items and this was used to avoid extreme responses, which turned out to be extremely useful in previous research [75]. Additionally, respondents had different response formats for expressing a reaction to the statements, such as the Likert scale and open-ended questions. During data collection, we collected responses from employees who worked in tourism entities located in different sites. Further, we tried to increase honest responses by explaining in the cover letter that respondents’ full anonymity will be guaranteed, that there were no correct or incorrect answers, and that we, as researchers, will only have access to their individual responses. Reducing the impact of CMV was possible by ensuring participant anonymity and ensuring that they know that the data will be protected and used for adequate purposes [76,77]. Managers of hotels, restaurants, and travel agencies handled the online distribution process of the questionnaires to their employees, so this might affect the bias of the selection procedure. Future studies should try to collect data from hospitality employees directly as much as possible. To decrease the chances of CMV, future research should apply some other procedural remedies, such as collecting the data at different points in time, using different media (social networks), or trying to change the questions’ order to control for bias effects. Since the impact of CMV sometimes cannot be fully removed by using procedural remedies, we applied some statistical analysis—a Harman one-factor analysis. In future research, it is advisable to apply other statistical techniques that not only detect CMV, but can assess the nature and the magnitude of the bias. One of them could be the directly measured latent factor method. 

Further, personal resources were viewed as a composite element, but since there is no explicit agreement in the literature on whether to use them as one construct or to analyze the influence of each separately, future research should investigate both solutions. Additionally, we used self-efficacy, hope, resilience, and optimism as the indicators of personal resources, but future research may explore the role of other personal resources, such as positive affectivity and intrinsic motivation. The current study used employees’ well-being at work as the only outcome, but future research can take into consideration other outcomes, such as work engagement, turnover intentions, or extra-role customer service behavior. Regarding the demographic characteristics, we included only three (gender, marital status, and years of work experience) but, for a better understanding of employees’ well-being at work, it would be desirable to determine whether there are differences between employees in the hospitality sector depending on their age, position in the organization (managers and non-managers), education, and income. Finally, our study included hospitality employees only from Serbia, so making general conclusions about employees’ well-being at work can be difficult. Doing similar research in other countries with different cultural, historical, and economic backgrounds and defining which organizational interventions will most likely increase well-being at work will significantly improve the understanding of the problem and make a significant theoretical contribution. 

## 5. Conclusions

Using data derived from frontline employees of the Serbian hospitality industry, the study investigated whether different elements of the work environment (manifested by job demands, job control, and social support) and personal resources were linked to employees’ well-being at work, and if personal resources can have a moderating role in the relationship between the work environment and well-being at work. Self-efficacy, hope, optimism, and resilience were selected as the main components of personal resources. Generally, there was empirical support for almost all hypotheses, except for the indirect effect of personal resources on the relationship between social support and well-being at work. The results of the study showed that the level of the work environment components is a significant predictor of employees’ well-being at work, especially job demands and social support, indicating that high job demands and long-term exposure to them cause negative emotions toward the job, while colleagues’ and supervisors’ support enhance positive feeling toward work-related tasks. Further, personal resources help employees to easier handle job demands. The findings highlighted that the work environment can have a crucial impact on employees’ well-being at work, so redesigning some of its components (such as the pace and volume of job demands and types of social support) will be important for developing positive emotions toward the job. On the other side, the results pointed to the fact that it is highly recommended to invest in the employees’ personal resources since their importance for their well-being at work was evident. 

## Figures and Tables

**Figure 1 ijerph-19-16165-f001:**
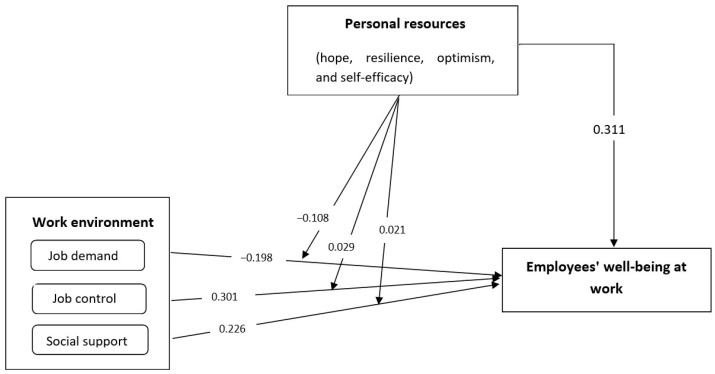
The results of the path model are based on standardized regression weights.

**Figure 2 ijerph-19-16165-f002:**
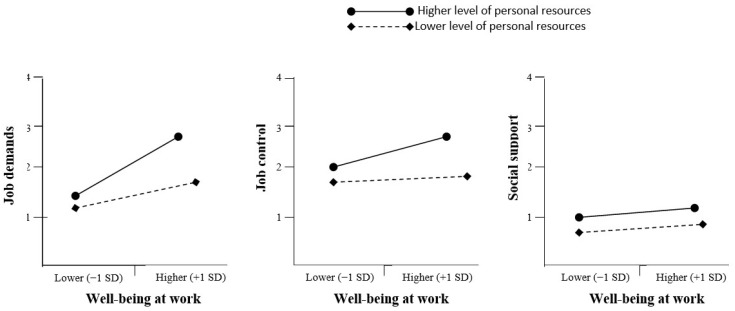
Moderating effect of personal resources.

**Table 1 ijerph-19-16165-t001:** Descriptive statistics and correlations among the variables.

	Mean	Std.	1	2	3	4	5	6	7	8
Gender	1.50	0.50								
2. Marital status	2.14	1.17	0.08							
3. Years of work experience	3.68	1.98	0.06	0.05						
4. Job demands	3.14	3.18	0.05 **	0.27 **	0.20 **	(0.74)				
5. Job control	3.36	2.21	0.08 **	0.08 **	0.16 **	0.33	(0.68)			
6. Social support	3.92	2.06	0.07 **	0.11 **	0.19 **	0.26	0.29	(0.82)		
7. Personal resources	4.01	1.31	0.06	0.06	0.17	−0.13 **	0.28	0.11 **	(0.77)	
8. Well-being at work	3.42	0.87	0.04 **	0.14 **	0.13 **	−0.15 **	0.04 **	0.31 **	0.37 **	(0.83)

Note. Scale reliability estimates are presented diagonally, in parentheses; ** *p* < 0.001.

**Table 2 ijerph-19-16165-t002:** The results of the model (standardized regression weights).

Hypothesized Paths	β	*t*-Value	*p*-Value	RMSEA (95% CI) *	Effect Size	Decision
Direct hypotheses						
H1a: Job demands → Well-being at work	−0.198	6.132	0.001	(−0.314, −0.175)	0.048	Supported
H1b: Job control → Well-being at work	0.301	5.704	0.001	(0.148, 0.322)	0.059	Supported
H1c: Social support → Well-being at work	0.226	4.386	0.001	(0.131, 0.247)	0.041	Supported
H2: Personal resources → Well-being at work	0.311	4.369	0.001	(0.142, 0.357)	0.044	Supported
Indirect hypotheses (Moderating effects)						
H3a: Job demands → Personal resources → Well-being at work	−0.108	4.551	0.001	(−0.175, −0.041)		Supported
H3b: Job control → Personal resources → Well-being at work	0.029	6.483	0.001	(0.024, 0.119)		Supported
H3c: Social support → Personal resources → Well-being at work	0.021	5.702	0.09	(0.010, 0.063)		Not supported

Goodness-of-fit statistics for the structural model: *χ*^2^ = 2543.294, *df* = 1279, *p* < 0.001, *χ*^2^/*df* = 2.875, RMSEA = 0.046, CFI = 0.919, IFI = 0.95, TLI = 0.903; * RMSEA = root mean squared error of approximation with a 95% confidence interval.

## Data Availability

Datasets are available upon request to the corresponding author.

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
