# Peer review of "Significance of the Work Environment and Personal Resources for Employees’ Well-Being at Work in the Hospitality Sector"

_ijerph, 2022, doi:10.3390/ijerph192316165_

Round 1

Reviewer 1 Report

Dear Authors

In advance, congratulate you for supporting the improvement of organizations and the relationship with people. It has good data and the structural model that it presents contributes significantly to the scientific process and therefore helps to validate some hypotheses.

I respectfully suggest that the topic of writing be improved so that, in addition to generating interest in scientific reading, it is an element of value for organizations in the sector that you have investigated.

Sometimes science leans too much towards technical concepts and for this reason the productive or real sector does not value our research.

For this reason, we invite you to the conclusions and the results will provide reading interest so that organizations can make decisions based on what is stated by the authors.

Author Response

We thank the reviewer for his/her professional and positive attitude toward our research. We found your remarks very helpful in improving our article.

 In the results section, we gave short explanations of why we used some specific statistical analyses – correlation analysis and structural equation modeling. Using a simple explanation, we described the meaning of those two techniques and presented their advantages. 

Regarding the conclusions, we believe that subsection 4.1. (Practical and theoretical implications) could be very useful for the management of hospitality organizations. In this subsection, we recommended some guidelines according to the results we got. We tried to provide specific proposals, for example: “…to decrease negative effects of high job demands, hospitality managers should set the mutual expectation and clear priorities with their employees, make sure that employees are at the right workplaces according to their abilities and interests, enables task or even job rotation…”, or “…organize work with clear goals and reasonable timeframe for the completion of work-related tasks, interact with employees by giving them feedback and being specific, showing empathy and making the conversation "a two-way street"…”. In this subsection, we especially tried to avoid any technical concepts or complex (statistical) terms so that they would be useful to people working in practice. 

Reviewer 2 Report

Dear authors:

I have reviewed the study ““Significance of Work Environment and Personal Resources for

Employees' Well-being at Work in Hospitality Sector”. The topic is interesting and adds empirical evidence to JD-R model. Nevertheless, some improvements are suggested. Please, I suggest rethinking the model (results). At the theoretical background is argued a mediation modulated model and at the end (results section), you test in the same model mediation and moderation, please rethink which is your research question and decide if personal resources are mediators and moderators.  

Introduction and theoretical framework

1. I recommend you that include references in the first paragraph (linen45). It is asserted “Gui

 reduced well-being can be expressed through appearance of work-related anx-44 iety, stress, depression, fatigue and other undesirable states and emotions” but this statement is not supported with any reference.

2. The hypothesis H2a H2b H2c are not enough justified, since it is needed to explain better the relationship between the variables that you consider personal resources and the independent variables. If not there are previous findings, you could use JD-R model and add more rationale.

3. In p.5 line 205 it is written mediator, I think that you would like to write moderator. Additionally, the article of Tremblay and Messervey is not related with the moderator variable that are used in your MS. Could you exclude it?. ,

4. I recommend that rewrite the hypothesis showing that VI is related to VD, in the place that is correlated.

Results

1.     Correlation table does not include all the variables measures in the study.

2.     Please, include the measurement model of the main variables to show CMV, that there is no CMV

3.     ¿What was CI interval of RMSEA at the results?

4.     As the sample is cross-sectional, the variables only could influence one in another, since there is no causality. Please remove words such as affect or similar.

5.     Please, rewrite table 2 and differentiate between direct and indirect effects. Reading results, it seems that hypothesized path H1b is not supported. 95%CI. At least, make them more understandable. On the whole, review 95%CI.

6.     Use the B, β, and ak+ when it is necessary in the place of using simple slope high or estimate.

Discussion

I suggest you adding a discussion about the influence of the context of hospitality in this section. I recommend you that you explain CMV and how to solve or showed that it is not an issue in your sample.

Author Response

We're attaching the responses to the second reviewer's comments (check the Authors_report). 

Round 2

Reviewer 2 Report

Dear authors,

Congratulations for the changes that you have made in ““Significance of Work Environment and Personal Resources for  Employees' Well-being at Work in Hospitality Sector”. The article has considerably improved.

 Some minor issues:

 CMV is considered a limitation of the study, then it used to be discussed in limitation section.

 It is clear that you proposed a moderation, but why you related IV with the moderation if the moderation is an interaction between the independent variable and the moderating variable. Again, you could not have in the same model (as it shown in the figure) a mediated moderating effect with the same variable, as far as I know. Please, justify this model. 

 I could not follow how authors made SEM. Did you use SEM or path analysis method? If you used SEM, it is needed to change the figure and add the items.

 Anyway, in moderation analysis is used unstandardized  slopes, so you must use B.

 Review table of results , since unsupported results has a significant 95%CI interval, it could be a typo.

At last, could you include p values in the correlations’ table and add RMSEA 95% CI interval in the results. Please, could you eliminate 95%CI interval line 405 page9, since it has nonsense. 

Author Response

Dear reviewer,

Thank you for your helpful suggestions. We're attaching the responses to your comments.

Kindly,

Authors of the paper
